# Safe Reinforcement Learning by Imagining the Near Future

**Garrett Thomas**
Stanford University
gwthomas@stanford.edu

**Yuping Luo**
Princeton University
yupingl@cs.princeton.edu

**Tengyu Ma**
Stanford University
tengyuma@stanford.edu

## Abstract

Safe reinforcement learning is a promising path toward applying reinforcement learning algorithms to real-world problems, where suboptimal behaviors may lead to actual negative consequences. In this work, we focus on the setting where unsafe states can be avoided by planning ahead a short time into the future. In this setting, a model-based agent with a sufficiently accurate model can avoid unsafe states. We devise a model-based algorithm that heavily penalizes unsafe trajectories, and derive guarantees that our algorithm can avoid unsafe states under certain assumptions. Experiments demonstrate that our algorithm can achieve competitive rewards with fewer safety violations in several continuous control tasks.

## 1 Introduction

Reinforcement learning (RL) enables the discovery of effective policies for sequential decision-making tasks via trial and error [Mnih et al., 2015, Gu et al., 2016, Bellemare et al., 2020]. However, in domains such as robotics, healthcare, and autonomous driving, certain kinds of mistakes pose danger to people and/or objects in the environment. Hence there is an emphasis on the safety of the policy, both at execution time and while interacting with the environment during learning. This issue, referred to as *safe exploration*, is considered an important problem in AI safety [Amodei et al., 2016].

In this work, we advocate a model-based approach to safety, meaning that we estimate the dynamics of the system to be controlled and use the model for planning (or more accurately, policy improvement). The primary motivation for this is that a model-based method has the potential to anticipate safety violations *before they occur*. Often in real-world applications, the engineer has an idea of what states should be considered violations of safety: for example, a robot colliding rapidly with itself or surrounding objects, a car driving on the wrong side of the road, or a patient's blood glucose levels spiking.Yet model-free algorithms typically lack the ability to incorporate such prior knowledge and must encounter some safety violations before learning to avoid them.

We begin with the premise that in practice, forward prediction for relatively few timesteps is sufficient to avoid safety violations. Consider the illustrative example in Figure 1, in which an agent controls the acceleration (and thereby, speed) of a car by pressing the gas or brake (or nothing). Note that there is an upper bound on how far into the future the agent would have to plan to foresee and (if possible) avoid any collision, namely, the amount of time it takes to bring the car to a complete stop.

Assuming that the horizon required for detecting unsafe situations is not too large, we show how to construct a reward function with the property that an optimal policy will never incur a safety violation. A short prediction horizon is also beneficial for model-based RL, as the well-known issue of *compounding error* plagues long-horizon prediction [Asadi et al., 2019]: imperfect predictions are fed back into the model as inputs (possibly outside the distribution of inputs in the training data), leading to progressively worse accuracy as the prediction horizon increases.

35th Conference on Neural Information Processing Systems (NeurIPS 2021).

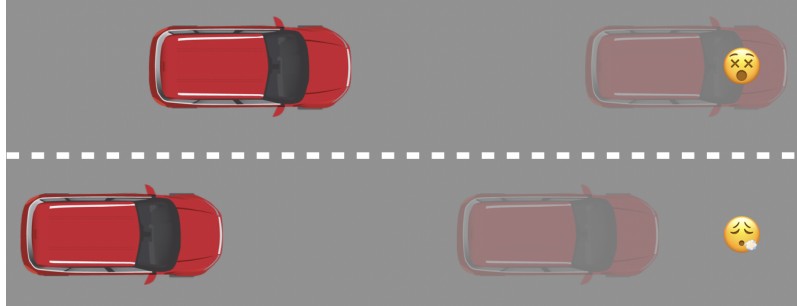

Figure 1: An illustrative example. The agent controls the speed of a car by pressing the accelerator or brake (or neither), attempting to avoid any obstacles such as other cars or people in the road. The top car has not yet come into contact with the pedestrian, but cannot avoid the pedestrian from its current position and speed, even if it brakes immediately. The bottom car can slow down before hitting the pedestrian. If the bottom car plans several steps into the future, it could reduce its speed to avoid the "irrecoverable" situation faced by the top car.

Our main contribution is a model-based algorithm that utilizes a reward penalty – the value of which is prescribed by our theoretical analysis – to guarantee safety (under some assumptions). Experiments indicate that the practical instantiation of our algorithm, Safe Model-Based Policy Optimization (SMBPO), effectively reduces the number of safety violations on several continuous control tasks, achieving a comparable performance with far fewer safety violations compared to several model-free safe RL algorithms. Code is made available at https://github.com/gwthomas/Safe-MBPO.

## 2 Background

In this work, we consider a deterministic[1] Markov decision process (MDP) $M = (\mathcal{S}, \mathcal{A}, T, r, \gamma)$, where $\mathcal{S}$ is the state space, $\mathcal{A}$ the action space, $T : \mathcal{S} \times \mathcal{A} \to \mathcal{S}$ the transition dynamics, $r : \mathcal{S} \times \mathcal{A} \to [r_{\min}, r_{\max}]$ the reward function, and $\gamma \in [0, 1)$ the discount factor. A policy $\pi : \mathcal{S} \to \Delta(\mathcal{A})$ determines what action to take at each state. A trajectory is a sequence of states and actions $\tau = (s_0, a_0, r_0, s_1, a_1, r_1, \dots)$ where $s_{t+1} = T(s_t, a_t)$ and $r_t = r(s_t, a_t)$.

Typically, the goal is to find a policy which maximizes the expected discounted return $\eta(\pi) = \mathbb{E}^\pi[\sum_{t=0}^\infty \gamma^t r_t]$. The notation $\mathbb{E}^\pi$ denotes that actions are sampled according to $a_t \sim \pi(s_t)$. The initial state $s_0$ is drawn from an initial distribution which we assume to be fixed and leave out of the notation for simplicity.

The $Q$ function $Q^\pi(s, a) = \mathbb{E}^\pi[\sum_{t=0}^\infty \gamma^t r_t \mid s_0 = s, a_0 = a]$ quantifies the conditional performance of a policy $\pi$ assuming it starts in a specific state $s$ and takes action $a$, and the value function $V^\pi(s) = \mathbb{E}_{a \sim \pi(s)}[Q^\pi(s, a)]$ averages this quantity over actions. The values of the best possible policies are denoted $Q^*(s, a) = \max_\pi Q^\pi(s, a)$ and $V^*(s) = \max_\pi V^\pi(s)$. The function $Q^*$ has the important property that any optimal policy $\pi^* \in \arg\max_\pi \eta(\pi)$ must satisfy $\mathbb{P}(a^* \in \arg\max_a Q^*(s, a)) = 1$ for all states $s$ and actions $a^* \sim \pi^*(s)$. $Q^*$ is the unique fixed point of the Bellman operator

$$\mathcal{B}^* Q(s, a) = r(s, a) + \gamma \max_{a'} Q(s', a') \quad \text{where } s' = T(s, a) \tag{1}$$

In model-based RL, the algorithm estimates a dynamics model $\widehat{T}$ using the data observed so far, then uses the model for planning or data augmentation. The theoretical justification for model-based RL is typically based some version of the "simulation lemma", which roughly states that if $\widehat{T} \approx T$ then $\hat{\eta}(\pi) \approx \eta(\pi)$ [Kearns and Singh, 2002, Luo et al., 2018].

---

[1]Determinism makes safety essentially trivial in tabular MDPs. We focus on tasks with continuous state and/or action spaces. See Appendix A.2 for a possible extension of our approach to stochastic dynamics.

## 3 Method

In this work, we train safe policies by modifying the reward function to penalize safety violations. We assume that the engineer specifies $\mathcal{S}_{\text{unsafe}}$, the set of states which are considered safety violations.

We must also account for the existence of states which are not themselves unsafe, but lead inevitably to unsafe states regardless of what actions are taken.

**Definition 3.1.** *A state $s$ is said to be*

- *a **safety violation** if $s \in \mathcal{S}_{\text{unsafe}}$.*

- ***irrecoverable** if $s \notin \mathcal{S}_{\text{unsafe}}$ but for any sequence of actions $a_0, a_1, a_2, \ldots$, the trajectory defined by $s_0 = s$ and $s_{t+1} = T(s_t, a_t)$ for all $t \in \mathbb{N}$ satisfies $s_{\bar{t}} \in \mathcal{S}_{\text{unsafe}}$ for some $\bar{t} \in \mathbb{N}$.*

- ***unsafe** if it is unsafe or irrecoverable, or **safe** otherwise.*

We remark that these definitions are similar to those introduced in prior work on safe RL [Hans et al., 2008]. Crucially, we do not assume that the engineer specifies which states are (ir)recoverable, as that would require knowledge of the system dynamics. However, we do assume that a safety violation must come fairly soon after entering an irrecoverable region:

**Assumption 3.1.** *There exists a horizon $H^* \in \mathbb{N}$ such that, for any irrecoverable states $s$, any sequence of actions $a_0, \ldots, a_{H^*-1}$ will lead to an unsafe state. That is, if $s_0 = s$ and $s_{t+1} = T(s_t, a_t)$ for all $t \in \{0, \ldots, H^* - 1\}$, then $s_{\bar{t}} \in \mathcal{S}_{\text{unsafe}}$ for some $\bar{t} \in \{1, \ldots, H^*\}$.*

This assumption rules out the possibility that a state leads inevitably to termination but takes an arbitrarily long time to do so. The implication of this assumption is that a perfect lookahead planner which considers the next $H$ steps into the future can avoid not only the unsafe states, but also any irrecoverable states, with some positive probability.

### 3.1 Reward penalty framework

Now we present a reward penalty framework for guaranteeing safety. Let $\widetilde{M}_C = (\mathcal{S}, \mathcal{A}, \widetilde{T}, \tilde{r}, \gamma)$ be an MDP with reward function and dynamics

$$\left( \tilde{r}(s, a), \widetilde{T}(s, a) \right) = \begin{cases} (r(s, a), T(s, a)) & s \notin \mathcal{S}_{\text{unsafe}} \\ (-C, s) & s \in \mathcal{S}_{\text{unsafe}} \end{cases} \tag{2}$$

where the terminal cost $C \in \mathbb{R}$ is a constant (more on this below). That is, unsafe states are "absorbing" in that they transition back into themselves and receive the reward of $-C$ regardless of what action is taken.

The basis of our approach is to determine how large $C$ must be so that the Q values of actions leading to unsafe states are less than the Q values of safe actions.

**Lemma 3.1.** *Suppose that Assumption 3.1 holds, and let*

$$C > \frac{r_{\max} - r_{\min}}{\gamma^{H^*}} - r_{\max}. \tag{3}$$

*Then for any state $s$, if $a$ is a safe action (i.e. $T(s, a)$ is a safe state) and $a'$ is an unsafe action (i.e. $T(s, a)$ is unsafe), it holds that $\widetilde{Q}^*(s, a) > \widetilde{Q}^*(s, a')$, where $\widetilde{Q}^*$ is the $Q^*$ function for the MDP $\widetilde{M}_C$.*

*Proof.* Since $a'$ is unsafe, it leads to an unsafe state in at most $H^*$ steps by assumption. Thus the discounted reward obtained is at most

$$\sum_{t=0}^{H^*-1} \gamma^t r_{\max} + \sum_{t=H^*}^{\infty} \gamma^t(-C) = \frac{r_{\max}(1 - \gamma^{H^*}) - C\gamma^{H^*}}{1 - \gamma} \tag{4}$$

By comparison, the safe action $a$ leads to another safe state, where it can be guaranteed to never encounter a safety violation. The reward of staying within the safe region forever must be at least $\frac{r_{\min}}{1-\gamma}$. Thus, it suffices to choose $C$ large enough that

$$\frac{r_{\max}(1 - \gamma^{H^*}) - C\gamma^{H^*}}{1 - \gamma} < \frac{r_{\min}}{1 - \gamma} \tag{5}$$

Rearranging, we arrive at the condition stated. $\qquad \square$

The important consequence of this result is that an optimal policy for this MDP $\widetilde{M}$ will always take safe actions. However, in practice we cannot compute $\widetilde{Q}^*$ without knowing the dynamics model $T$. Therefore we extend our result to the model-based setting where the dynamics are imperfect.

## 3.2 Extension to model-based rollouts

We prove safety for the following theoretical setup. Suppose we have a dynamics model that outputs *sets* of states $\widehat{T}(s, a) \subseteq \mathcal{S}$ to account for uncertainty.

**Definition 3.2.** *We say that a set-valued dynamics model $\widehat{T} : \mathcal{S} \times \mathcal{A} \rightarrow \mathcal{P}(\mathcal{S})$[2] is **calibrated** if $T(s, a) \in \widehat{T}(s, a)$ for all $(s, a) \in \mathcal{S} \times \mathcal{A}$.*

We define the *Bellmin* operator:

$$\underline{\mathcal{B}}^* Q(s, a) = \tilde{r}(s, a) + \gamma \min_{s' \in \widehat{T}(s,a)} \max_{a'} Q(s', a') \tag{6}$$

**Lemma 3.2.** *The Bellmin operator $\underline{\mathcal{B}}^*$ is a $\gamma$-contraction in the $\infty$-norm.*

The proof is deferred to Appendix A.1. As a consequence Lemma 3.2 and Banach's fixed-point theorem, $\underline{\mathcal{B}}^*$ has a unique fixed point $\underline{Q}^*$ which can be obtained by iteration. This fixed point is a lower bound on the true Q function if the model is calibrated:

**Lemma 3.3.** *If $\widehat{T}$ is calibrated in the sense of Definition 3.2, then $\underline{Q}^*(s, a) \le \widetilde{Q}^*(s, a)$ for all $(s, a)$.*

*Proof.* Let $\tilde{\mathcal{B}}^*$ denote the Bellman operator with reward function $\tilde{r}$. First, observe that for any $Q, Q' : \mathcal{S} \times \mathcal{A} \rightarrow \mathbb{R}$, $Q \le Q'$ pointwise implies $\underline{\mathcal{B}}^* Q \le \mathcal{B}^* Q'$ pointwise because we have $\tilde{r}(s, a) + \gamma \max_{a'} Q(s', a') \le \tilde{r}(s, a) + \gamma \max_{a'} Q'(s', a')$ pointwise and the min defining $\underline{\mathcal{B}}^*$ includes the true $s' = T(s, a)$.

Now let $Q_0$ be any inital Q function. Define $\widetilde{Q}_k = (\tilde{\mathcal{B}}^*)^k Q_0$ and $\underline{Q}_k = (\underline{\mathcal{B}}^*)^k Q_0$. An inductive argument coupled with the previous observation shows that $\underline{Q}_k \le \widetilde{Q}_k$ pointwise for all $k \in \mathbb{N}$. Hence, taking the limits $\widetilde{Q}^* = \lim_{k \to \infty} \widetilde{Q}_k$ and $\underline{Q}^* = \lim_{k \to \infty} \underline{Q}_k$, we obtain $\underline{Q}^* \le \widetilde{Q}^*$ pointwise. $\square$

Now we are ready to present our main theoretical result.

**Theorem 3.1.** *Let $\widehat{T}$ be a calibrated dynamics model and $\pi^*(s) = \arg\max_a \underline{Q}^*(s, a)$ the greedy policy with respect to $\underline{Q}^*$. Assume that Assumption 3.1 holds. Then for any $s \in \mathcal{S}$, if there exists an action $a$ such that $\underline{Q}^*(s, a) \ge \frac{r_{\min}}{1-\gamma}$, then $\pi^*(s)$ is a safe action.*

*Proof.* Lemma 3.2 implies that $\underline{Q}^*(s, a) \le \widetilde{Q}^*(s, a)$ for all $(s, a) \in \mathcal{S} \times \mathcal{A}$.

As shown in the proof of Lemma 3.1, any unsafe action $a'$ satisfies

$$\underline{Q}^*(s, a') \le \widetilde{Q}^*(s, a') \le \frac{r_{\max}(1 - \gamma^{H^*}) - C\gamma^{H^*}}{1 - \gamma} \tag{7}$$

Similarly if $\underline{Q}^*(s, a) \ge \frac{r_{\min}}{1-\gamma}$, we also have

$$\frac{r_{\min}}{1 - \gamma} \le \underline{Q}^*(s, a) \le \widetilde{Q}^*(s, a) \tag{8}$$

so $a$ is a safe action. Taking $C$ as in inequality (3) guarantees that $\underline{Q}^*(s, a) > \underline{Q}^*(s, a')$, so the greedy policy $\pi^*$ will choose $a$ over $a'$. $\square$

This theorem gives us a way to establish safety using only short-horizon predictions. The conclusion conditionally holds for any state $s$, but for $s$ far from the observed states, we expect that $\widehat{T}(s, a)$ likely has to contain many states in order to satisfy the assumption that it contains the true next state, so that $\underline{Q}^*(s, a)$ will be very small and we may not have any action such that $\underline{Q}^*(s, a) \ge \frac{r_{\min}}{1-\gamma}$. However, it is plausible to believe that there can be such an $a$ for the set of states in the replay buffer, $\{s : (s, a, r, s') \in \mathcal{D}\}$.

---

[2] $\mathcal{P}(X)$ is the powerset of a set $X$.

---

**Algorithm 1** Safe Model-Based Policy Optimization (SMBPO)

---

**Require:** Horizon $H$
1: Initialize empty buffers $\mathcal{D}$ and $\widehat{\mathcal{D}}$, an ensemble of probabilistic dynamics $\{\widehat{T}_{\theta_i}\}_{i=1}^N$, policy $\pi_\phi$, critic $Q_\psi$.
2: Collect initial data using random policy, add to $\mathcal{D}$.
3: **for** episode $1, 2, \ldots$ **do**
4:     Collect episode using $\pi_\phi$; add the samples to $\mathcal{D}$. Let $\ell$ be the length of the episode.
5:     Re-fit models $\{\widehat{T}_{\theta_i}\}_{i=1}^N$ by several epochs of SGD on $L_{\widehat{T}}(\theta_i)$ defined in (9)
6:     Compute empirical $r_{\min}$ and $r_{\max}$, and update $C$ according to (3).
7:     **for** $\ell$ times **do**
8:         **for** $n_{\text{rollout}}$ times (in parallel) **do**
9:             Sample $s \sim \mathcal{D}$.
10:             Startin from $s$, roll out $H$ steps using $\pi_\phi$ and $\{\widehat{T}_{\theta_i}\}$; add the samples to $\widehat{\mathcal{D}}$.
11:         **for** $n_{\text{actor}}$ times **do**
12:             Draw samples from $\mathcal{D} \cup \widehat{\mathcal{D}}$.
13:             Update $Q_\psi$ by SGD on $L_Q(\psi)$ defined in (10) and target parameters $\bar{\psi}$ according to (12).
14:             Update $\pi_\phi$ by SGD on $L_\pi(\phi)$ defined in (13).

---

### 3.3 Practical algorithm

Based (mostly) on the framework described in the previous section, we develop a deep model-based RL algorithm. We build on practices established in previous deep model-based algorithms, particularly MBPO [Janner et al., 2019] a state-of-the-art model-based algorithm (which does not emphasize safety).

The algorithm, dubbed **Safe Model-Based Policy Optimization (SMBPO)**, is described in Algorithm 1. It follows a common pattern used by online model-based algorithms: alternate between collecting data, re-fitting the dynamics models, and improving the policy.

Following prior work [Chua et al., 2018, Janner et al., 2019], we employ an ensemble of (diagonal) Gaussian dynamics models $\{\widehat{T}_{\theta_i}\}_{i=1}^N$, where $\widehat{T}_i(s, a) = \mathcal{N}(\mu_{\theta_i}(s, a), \text{diag}(\sigma_{\theta_i}^2(s, a)))$, in an attempt to capture both aleatoric and epistemic uncertainties. Each model is trained via maximum likelihood on all the data observed so far:

$$L_{\widehat{T}}(\theta_i) = -\mathbb{E}_{(s,a,r,s') \sim \mathcal{D}} \log \widehat{T}_{\theta_i}(s', r \,|\, s, a) \tag{9}$$

However, random differences in initialization and mini-batch order while training lead to different models. The model ensemble can be used to generate uncertainty-aware predictions. For example, a set-valued prediction can be computed using the means $\widehat{T}(s, a) = \{\mu_{\theta_i}(s, a)\}_{i=1}^N$.

The models are used to generate additional samples for fitting the $Q$ function and updating the policy. In MBPO, this takes the form of short model-based rollouts, starting from states in $\mathcal{D}$, to reduce the risk of compounding error. At each step in the rollout, a model $\widehat{T}_i$ is randomly chosen from the ensemble and used to predict the next state. The rollout horizon $H$ is chosen as a hyperparameter, and ideally exceeds the (unknown) $H^*$ from Assumption 3.1. In principle, one can simply increase $H$ to ensure it is large enough, but this increases the opportunity for compounding error.

MBPO is based on the soft actor-critic (SAC) algorithm, a widely used off-policy maximum-entropy actor-critic algorithm [Haarnoja et al., 2018a]. The $Q$ function is updated by taking one or more SGD steps on the objective

$$L_Q(\psi) = \mathbb{E}_{(s,a,r,s') \sim \mathcal{D} \cup \widehat{\mathcal{D}}}[(Q_\psi(s, a) - (r + \gamma V_{\bar{\psi}}(s'))^2] \tag{10}$$

$$\text{where} \quad V_{\bar{\psi}}(s') = \begin{cases} -C/(1 - \gamma) & s' \in \mathcal{S}_{\text{unsafe}} \\ \mathbb{E}_{a' \sim \pi(s')}[Q_{\bar{\psi}}(s', a') - \alpha \log \pi_\phi(a' \,|\, s')] & s' \notin \mathcal{S}_{\text{unsafe}} \end{cases} \tag{11}$$

The scalar $\alpha$ is a hyperparameter of SAC which controls the tradeoff between entropy and reward. We tune $\alpha$ using the procedure suggested by Haarnoja et al. [2018b].

The $\bar{\psi}$ are parameters of a "target" $Q$ function which is updated via an exponential moving average towards $\psi$:

$$\bar{\psi} \leftarrow \tau\psi + (1 - \tau)\bar{\psi} \tag{12}$$

for a hyperparameter $\tau \in (0, 1)$ which is often chosen small, e.g., 0.005. This is a common practice used to promote stability in deep RL, originating from Lillicrap et al. [2015]. We also employ the

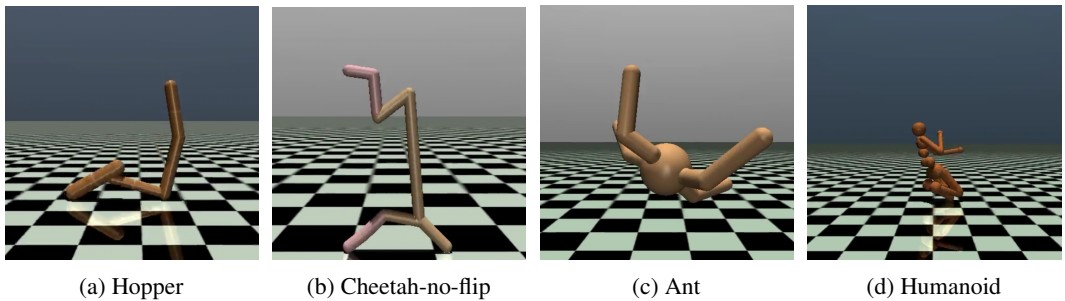

| (a) Hopper | (b) Cheetah-no-flip | (c) Ant | (d) Humanoid |

Figure 2: We show examples of failure states for the control tasks considered in experiments.

clipped double-Q method [Fujimoto et al., 2018] in which two copies of the parameters ($\psi_1$ and $\psi_2$) and target parameters ($\bar{\psi}_1$ and $\bar{\psi}_2$) are maintained, and the target value in equation (11) is computed using $\min_{i=1,2} Q_{\bar{\psi}_i}(s', a')$.

Note that in (10), we are fitting to the average TD target across models, rather than the min, even though we proved Theorem 3.2 using the Bellmin operator. We found that taking the average worked better empirically, likely because the min was overly conservative and harmed exploration.

The policy is updated by taking one or more steps to minimize

$$L_\pi(\phi) = \mathbb{E}_{s \sim \mathcal{D} \cup \widehat{\mathcal{D}}, a \sim \pi_\phi(s)}[\alpha \log \pi_\phi(a \,|\, s) - Q_\psi(s, a)]. \tag{13}$$

## 4 Experiments

In the experimental evaluation, we compare our algorithm to several model-free safe RL algorithms, as well as MBPO, on various continuous control tasks based on the MuJoCo simulator [Todorov et al., 2012]. Additional experimental details, including hyperparameter selection, are given in Appendix A.3.

### 4.1 Tasks

The tasks are described below:

- **Hopper**: Standard hopper environment from OpenAI Gym, except with the "alive bonus" (a constant) removed from the reward so that the task reward does not implicitly encode the safety objective. The safety condition is the usual termination condition for this task, which corresponds to the robot falling over.
- **Cheetah-no-flip**: The standard half-cheetah environment from OpenAI Gym, with a safety condition: the robot's head should not come into contact with the ground.
- **Ant**, **Humanoid**: Standard ant and humanoid environments from OpenAI Gym, except with the alive bonuses removed, and contact forces removed from the observation (as these are difficut to model). The safety condition is the usual termination condition for this task, which corresponds to the robot falling over.

For all of the tasks, the reward corresponds to positive movement along the $x$-axis (minus some small cost on action magnitude), and safety violations cause the current episode to terminate. See Figure 2 for visualizations of the termination conditions.

### 4.2 Algorithms

We compare against the following algorithms:

- **MBPO**: Corresponds to SMBPO with $C = 0$.
- **MBPO+bonus**: The same as MBPO, except adding back in the alive bonus which was subtracted out of the reward.

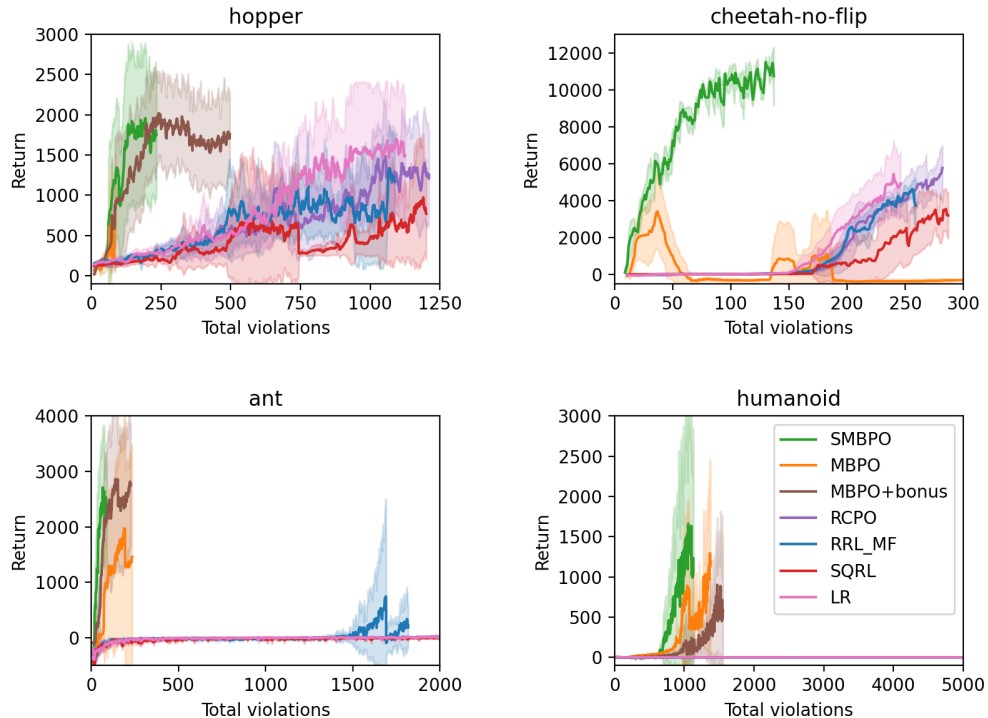

Figure 3: Undiscounted return of policy vs. total safety violations. We run 5 seeds for each algorithm independently and average the results. The curves indicate mean of different seeds and the shaded areas indicate one standard deviation centered at the mean.

- **Recovery RL, model-free (RRL-MF)**: Trains a critic to estimate the safety separately from the reward, as well as a recovery policy which is invoked when the critic predicts risk of a safety violation.
- **Lagrangian relaxation (LR)**: Forms a Lagrangian to implement a constraint on the risk, updating the dual variable via dual gradient descent.
- **Safety Q-functions for RL (SQRL)**: Also formulates a Lagrangian relaxation, and uses a filter to reject actions which are too risky according to the safety critic.
- **Reward constrained policy optimization (RCPO)**: Uses policy gradient to optimize a reward function which is penalized according to the safety critic.

All of the above algorithms except for MBPO are as implemented in the Recovery RL paper [Thananjeyan et al., 2020] and its publicly available codebase[3]. We follow the hyperparameter tuning procedure described in their paper; see Appendix A.3 for more details. A recent work [Bharadhwaj et al., 2020] can also serve as a baseline but the code has not been released.

Our algorithm requires very little hyperparameter tuning. We use $\gamma = 0.99$ in all experiments. We tried both $H = 5$ and $H = 10$ and found that $H = 10$ works slightly better, so we use $H = 10$ in all experiments.

### 4.3 Results

The main criterion in which we are interested is performance (return) vs. the cumulative number of safety violations. The results are plotted in Figure 3. We see that our algorithm performs favorably compared to model-free alternatives in terms of this tradeoff, achieving similar or better performance with a fraction of the violations.

---

[3]https://github.com/abalakrishna123/recovery-rl

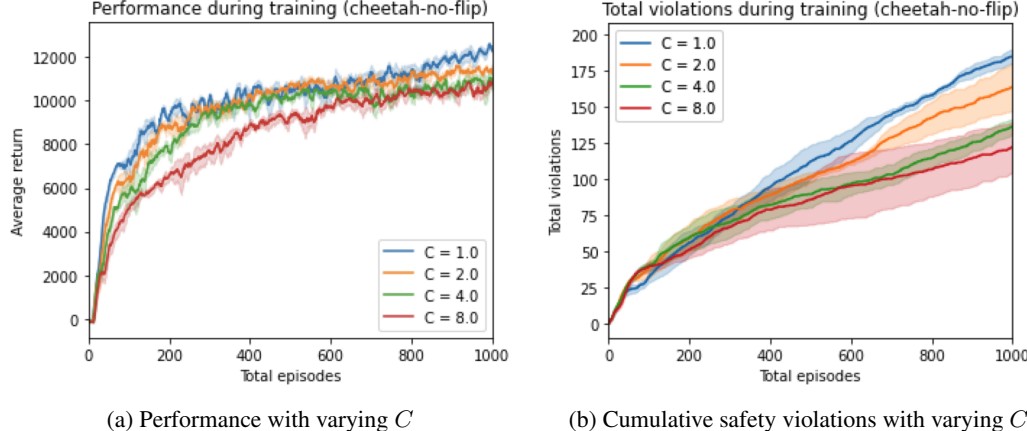

(a) Performance with varying $C$          (b) Cumulative safety violations with varying $C$

Figure 4: Increasing the terminal cost $C$ makes exploration more conservative, leading to fewer safety violations but potentially harming performance. This is what we expected: A larger $C$ focuses more on the safety requirement and learns more conservatively. Note that an epoch corresponds to 1000 samples.

MBPO is competitive in terms of sample efficiency but incurs more safety violations because it isn't designed explicitly to avoid them.

We also show in Figure 4 that hard-coding the value of $C$ leads to an intuitive tradeoff between performance and safety violations. With a larger $C$, SMBPO incurs substantially fewer safety violations, although the total rewards are learned slower.

## 5 Related Work

**Safe Reinforcement Learning**    Many of the prior works correct the action locally, that is, changing the action when the action is detected to lead to an unsafe state. Dalal et al. [2018] linearizes the dynamics and adds a layer on the top of the policy for correction. Bharadhwaj et al. [2020] uses rejection sampling to ensure the action meets the safety requirement. Thananjeyan et al. [2020] either trains a backup policy which is only used to guarantee safety, or uses model-predictive control (MPC) to find the best action sequence. MPC could also be applied in the short-horizon setting that we consider here, but it involves high runtime cost that may not be acceptable for real-time robotics control. Also, MPC only optimizes for rewards under the short horizon and can lead to suboptimal performance on tasks that require longer-term considerations [Tamar et al., 2017].

Other works aim to solve the constrained MDP more efficiently and better, with Lagrangian methods being applied widely. The Lagrangian multipliers can be a fixed hyperparameter, or adjusted by the algorithm [Tessler et al., 2018, Stooke et al., 2020]. The policy training might also have issues. The issue that the policy might change too fast so that it's no longer safe is addressed by building a trust region of policies [Achiam et al., 2017, Zanger et al., 2021] and further projecting to a safer policy [Yang et al., 2020], and another issue of too optimistic policy is addressed by Bharadhwaj et al. [2020] by using conservative policy updates. Expert information can greatly improve the training-time safety. Srinivasan et al. [2020], Thananjeyan et al. [2020] are provided offline data, while Turchetta et al. [2020] is provided *interventions* which are invoked at dangerous states and achieves zeros safety violations during training.

Returnability is also considered by Eysenbach et al. [2018] in practice, which trains a policy to return to the initial state, or by Roderick et al. [2021] in theory, which designs a PAC algorithm to train a policy without safety violations. Bansal et al. [2017] gives a brief overview of Hamilton-Jacobi Reachability and its recent progress.

**Model-based Reinforcement Learning**    Model-based reinforcement learning, which additionally learns the dynamics model, has gained its popularity due to its superior sample efficiency. Kurutach et al. [2018] uses an ensemble of models to produce imaginary samples to regularize leaerning and

reduce instability. The use of model ensemble is further explored by Chua et al. [2018], which studies different methods to sample trajectories from the model ensemble. Based on Chua et al. [2018], Wang and Ba [2019] combines policy networks with online learning. Luo et al. [2019] derives a lower bound of the policy in the real environment given its performance in the learned dynamics model, and then optimizes the lower bound stochastically. Our work is based on Janner et al. [2019], which shows the learned dynamics model doesn't generalize well for long horizon and proposes to use short model-generated rollouts instead of a full episodes. Dong et al. [2020] studies the expressivity of $Q$ function and model and shows that at some environments, the model is much easier to learn than the $Q$ function.

## 6  Conclusion

We consider the problem of safe exploration in reinforcement learning, where the goal is to discover a policy that maximizes the expected return, but additionally desire the training process to incur minimal safety violations. In this work, we assume access to a user-specified function which can be queried to determine whether or not a given state is safe. We have proposed a model-based algorithm that can exploit this information to anticipate safety violations before they happen and thereby avoid them. Our theoretical analysis shows that safety violations could be avoided with a sufficiently large penalty and accurate dynamics model. Empirically, our algorithm compares favorably to state-of-the-art model-free safe exploration methods in terms of the tradeoff between performance and total safety violations, and in terms of sample complexity.

## Acknowledgements

TM acknowledges support of Google Faculty Award, NSF IIS 2045685, the Sloan Fellowship, and JD.com. YL is supported by NSF, ONR, Simons Foundation, Schmidt Foundation, DARPA and SRC.

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
