# OpenReview forum: "Safe Reinforcement Learning by Imagining the Near Future"
_NeurIPS.cc/2021/Conference — NeurIPS 2021 Poster_

### Official Review · Reviewer_Fboi · 2021-07-13

**Rating:** 6
**Confidence:** 4

**Summary:**

The paper deals with one manifestation of safe exploration. Starting from a given safety function that classifies states as safe or unsafe, a modification of MBPO is used to change the Q-function in such a way that unsafe states should not be visited. The method is tested on two benchmarks.

**Limitations And Societal Impact:**

Yes

**Main Review:**

Strengths:\
The topic is relevant.\
The paper does not make exaggerated claims.

Weaknesses:\
Terms such as safe state and irrecoverable state are introduced, which were introduced very similarly already in [1], without mentioning that and citing [1].\
The experiments are based on only three replicates (3 random seeds).


Other comments:\
The expression "for any irrecoverable state s, any sequence of actions [...] will lead to an unsafe state" seems to assume deterministic MDPs. In general, stochastic MDPs it depends on random chance which states are visited. It should then rather be a) "will lead to an unsafe state with a probability greater than 0" or b) "will lead to an unsafe state with probability 1". Where a) is more in line with the concept of super-critical or irrecoverable states.\
Page 2, line 54: $\eta$ is used without having been introduced.

Typos / style:\
There is a missing reference on page 9, line 243\
„Startin“ -> „Starting“\
„the min“ -> „the minimum“\
„leaerning“ -> „learning“

[1] Alexander Hans et al., Safe Exploration for Reinforcement Learning. In Proceedings of the European Symposium on Artificial Neural Network, pages 143–148, 2008.

**After feedback** \
The authors addressed my concerns to my full satisfaction and I agree with the authors that in continuous state spaces even with a restriction to deterministic environments the topic is not trivial. Therefore, I think the paper is a valuable contribution and I increased the score to 6.


**Time Spent Reviewing:**

7

---

> ### Author Response · Authors · 2021-08-11
> **Response to reviewer Fboi**
>
> We thank the reviewer for referring us to the related paper of Hans et al. We will be sure to cite this work and mention their related definitions in the next revision of our paper. We will also run additional random seeds for the final results in the camera-ready, if accepted.
>
> It is true that we assume deterministic dynamics in the submission. We have since made progress on extending the theory to stochastic dynamics, see here: https://drive.google.com/file/d/1j69Xq4aKqFa-1WcIn5bQc8hugcDLD_bw/view?usp=sharing

---

> > ### Comment · Reviewer_Fboi · 2021-08-18
> > **All concerns addressed**
> >
> > The authors addressed my concerns to my full satisfaction.

---

> > > ### Author Response · Authors · 2021-08-23
> > > **Response to reviewer Fboi**
> > >
> > > We are happy to hear that your concerns were addressed! Thank you for your time reviewing our paper.

---

### Official Review · Reviewer_hu21 · 2021-07-14

**Rating:** 6
**Confidence:** 4

**Summary:**

This paper studies *safe* reinforcement learning (RL), in which an RL algorithm is tasked with learning a high-value behavioral policy while simultaneously minimizing the number of safety violations. To frame the work, the paper takes on a few key perspectives. First, that predicting only a few timesteps into the future is all that is needed to prevent safety violations. Second, that safety is defined in terms of the occupancy of set of states, as determined by the \textsc{Unsafe} predicate designed by an engineer. With this predicate in hand, a natural notion of an irrecoverable state is introduced, defined as those states that always lead to an unsafe state under any policy. Third, the paper assumes that environments are deterministic for the main analysis, and captures transition model uncertainty in the form of a next-state-set prediction.

With these perspectives in hand, the work establishes two primary contributions. The first is the formation of a safe proxy MDP, $\tilde{M}$, that uses a proxy reward and transition function. The proxy reward produces $-C$ for all unsafe states, where $C$ is a well chosen constant (to be returned to shortly). The proxy transition function replaces any unsafe transition with a self-loop. Hence, in reasoning with this proxy MDP, acting in an unsafe state will produce $-C$ reward indefinitely, thereby accumulating $C / (1-\gamma)$ value. By consequence, Prop. 1 illustrates that the Q values under the proxy MDP ensure safe actions always have higher values than unsafe actions. Then, section 3.2 takes this idea and extends it to the case when a model is being learned. In particular, it is assumed that the form of transition model learned is a "set-valued dynamics model" that maps each $(s,a)$ pair to a non-empty set of possible next states. With this definition of model, two Lemmas are introduced (1 and 2) that illustrate two things: First, that the cutely named "Bellmin" operator (which takes the s' achieved by minimizing Q w.r.t. possible next states in the set) is a $\gamma$-contraction (Lemma 1). Second, that a "calibrated" dynamics model (defined in Definition 2 as a set-valued model that always contains the true next-state in its next-state set) will ensure that the proxy Q function upper bounds the set-valued model's Q function. These two lemmas are used to Prove Theorem 1, which states that under a calibrated dynamics model, if there is an action with value greater than $r_{\min}/(1-\gamma)$, the greedy policy (w.r.t. $\underline{Q}$) will choose a safe action.

Then, following this analysis, the paper shifts focus on developing a practical, safe, model-based based algorithm (Algorithm 1, SMBPO). Experiments are conducted in two navigation tasks, hopper, and a cheetah variant, contrasting a variety of RL algorithms (many of which I was unfamiliar with). These include: MBPO, RRL-MF (recovery RL), LR (Lagrangian relaxation), SQRL (Safety Q-functions ), RCPO (Reward constrained policy optimization), and RSPO (Risk-sensitive policy optimization). The main quality explored here is the trade off each algorithm makes between performance and the number of safety violations.

**Limitations And Societal Impact:**

As stated, the most apparent limitation of the analysis is its restriction to deterministic transition models. I do not see how the core insights from Theorem 1 translate to the stochastic case, particularly due to the model of uncertainty chosen for the transition dynamics. A more suitable approach would be minimax over the worst case transition function, I suppose.

**Main Review:**


__Pros: Motivation, Clarity.__ There is a lot to like about this paper. The motivation is clearly articulated and easy to sympathize with: In order for RL agents to avoid safety violations while learning, it is sensible for agents to acquire a forward predictive model that can steer agent's behavior away from unsafe future states. This backdrop ideology frames much of the paper, which makes the definitions, method, and results fit very neatly with each other. I found definitions to be clear and succinct, as were proofs, assumptions, and most of the exposition in the paper. The decision to introduce irrecoverable states was quite sensible.

__Con: Modelling Choices.__ Some aspects of the technical setup, while quite clearly presented, were more difficult to get behind. In particular, I was surprised to see that the introduced definition of an MDP only allows for deterministic transition functions. This seems like a dramatic simplification, particularly in the case of safe RL. If an agent only ever needs to experience a transition once to learn it, why should we worry about the compounding error phenomenon? Moreover, we need not worry about adding extra pseudo-rewards to prevent agents from landing in irrecoverable states, as in order to experience the pseudo-reward the agent will need to take the action that moves it to the irrecoverable state already. Since the transitions are deterministic, this single experience is enough (with or without the pseudo-rewards) to allow the agent to avoid such decisions in the future. For this reason I found the setup of the analysis quite puzzling. Another consequence of this decision to be mindful of is that it does not force a trade-off between safety and value. That is, there is no notion of risk. In a stochastic environment we might imagine that good behavior might trade off with the potential for a safety violation (which do we as designers care more about? Is safety a hard or soft constraint?). I believe these questions are important to address, but due to deterministic transitions they can be avoided entirely as a safety violation is effectively assigned the lowest reward and that's that. Perhaps this is okay, as the work might build toward the general stochastic case, but I did not see any mention of the choice to stick with deterministic dynamics, nor suggestions for how to relaxat to stochastic dynamics in the analysis. This is what puzzled me the most about the paper. Moreover, the environments used in Section 4. (Experiments) are not themselves deterministic, so I found quite a start contrast between the framing from the previous sections and the experimental section.

__Writing Suggestions.__


1. Introduction
- "spiking.Yet" --> "spiking. Yet"

3. Method
- I found the definition of "irrecoverable" to be slightly confusing in the way that it is worded. I believe the phrase "for all $t \in \mathbb{N}$ satisfies $Unsafe(s_t)$ for some $t \in \mathbb{N}$". Having both quantifiers for some $t$ and for all $t$ is confusing. I see, on parsing this a few more times, it seems as though the first quantifier expresses the construction of the trajectory ($s_{t+1} = T(s_t, a_t)$), while the second quantifier expresses the existence of one unsafe state. I would suggest revising the wording here to make this more clear.
- Similarly, in assumption 1, it would be useful to explicitly state that $s_0$ is irrecoverable (this happens in the first sentence of the assumption, but not the second). So: "That is, if $s_0 = s$ is irrecoverable, then..."
- In Proposition 1, it is stated that "if $a$ is a safe action (i.e. $T(s,a)$ is a safe state)", but as I read it $T(s,a)$ should be a distribution over next states. Ah, I see, the transition function is actually defined to be deterministic. This seems to be quite a major limitation. I would have expected this to be emphasized early on: The compounding error problem cited at the beginning should not really be an issue for deterministic transitions, as the agent only needs to see a single transition per $(s,a)$ pair to learn the model.
- From Prop 1.'s proof, it turns out that $C$ is actually lower than the minimal reward. This also seems potentially odd: We reserve a particularly low quantity of reward for unsafe states. This seems okay, but it is likely worth calling attention to the fact that C is less than $r_{\min}$.

- Why consider model-based algorithms that output a set of states? Are you aware of any algorithms that actually do this, and are themselves designed for deterministic environments? Why would this be a valid way to represent uncertainty in a deterministic environment, since a single experience of a transition will fully resolve any uncertainty?

- Some of the references are off. Or, there is an odd convention at play that I found hard to follow. Definition 2 is labeled as such, but is referred to as Definition 3.2 (presumably since it appears in Section 3). I would suggest aligning these names. So either use \numberwithin to enforce "Definition 3.2" to appear, or when referring to the definition, use the listed number.


=============
== Update after rebuttal

In light of the author's rebuttal, I believe the focus on deterministic-but-continuous environments is indeed suitably interesting. The linked pdf also provides a partial path toward the stochastic setting (though I believe more work is needed to adequately expose and handle risk, exploration, and other considerations). In light of this, I believe the paper will be much improved if it takes time to highlight the nature of the setting, motivating why deterministic-but-continuous environments are an interesting case for safety (and bringing closer attention to the choice of deterministic continuous MDPs). Consequently, I am raising my score from 4 to 6.

**Time Spent Reviewing:**

6 hours

---

> ### Author Response · Authors · 2021-08-11
> **Response to reviewer hu21**
>
> We thank the reviewer for the many insightful comments! The main concern raised, which is that we assume deterministic dynamics, is indeed a limitation of the theory in the submission. However, this assumption does not make safety as trivial as the comments seem to suggest, considering that we allow for continuous state and action spaces. Although we only need to experience a transition once to know what happens for that $(s,a)$, in the continuous case there are infinitely many $(s,a)$ pairs and the agent is unlikely to experience the exact same transition again. Even if we know $(s,a)$ is safe, we cannot guarantee that a slightly different state and/or action would also be safe (without additional assumptions). For similar reasons, the issue of compounding error is still prevalent in deterministic dynamics when states/actions are continuous.
>
> Moreover, we have made progress on extending the theory to stochastic dynamics, please see here: https://drive.google.com/file/d/1j69Xq4aKqFa-1WcIn5bQc8hugcDLD_bw/view?usp=sharing
>
> Essentially, after modifying the definitions accordingly, the same strategy can be used to enforce that only sufficiently safe actions are chosen by the optimal policy.
>
> Below we respond to some specific comments from the review:
>
> *  "Moreover, the environments used in Section 4. (Experiments) are not themselves deterministic" // The environments’ transitions are deterministic. Only the initial state is randomized.
> * "From Prop 1.'s proof, it turns out that C is actually lower than the minimal reward. This also seems potentially odd: We reserve a particularly low quantity of reward for unsafe states. This seems okay, but it is likely worth calling attention to the fact that C is less than rmin." // This is true, and for the sake of clarity we will add additional commentary pointing out this fact to the reader, but we would argue it is fairly intuitive that $-C$ should be less than $r_{\min}$. If this were not the case, unsafe actions could have higher Q values than some low-reward safe actions, leading to potential safety violations.
> * "Why consider model-based algorithms that output a set of states? Are you aware of any algorithms that actually do this, and are themselves designed for deterministic environments?" // Set-valued prediction has long been used in statistics to represent uncertainty, e.g. confidence intervals/sets and conformal prediction. In deep model-based RL, it is common practice to employ ensembles of dynamics models to represent uncertainty (e.g. PETS, ME-TRPO, MBPO, ...) even when the dynamics are deterministic, and one can view the set of the models' predictions as a confidence set.
> * "Why would [outputting a set of states] be a valid way to represent uncertainty in a deterministic environment, since a single experience of a transition will fully resolve any uncertainty?" // It is true that under deterministic dynamics, any observed transition $(s,a,s')$ should lead to a confidence set containing only a single element. However, there remains the issue of how to predict for unseen states/actions. In continuous control problems where the state/action spaces admit a metric, one could use a Lipschitz assumption on the dynamics to produce a confidence set for states/actions which are close to the observed transitions.

---

> > ### Comment · Reviewer_hu21 · 2021-08-18
> > **Reply to authors**
> >
> > I thank the authors for their thoughtful response!
> >
> > I am sympathetic to the point that safety is still a concern in continuous-but-deterministic environments. I do believe, however, the paper would be improved by highlighting these characteristics throughout. That is, concretely, I suggest (1) Clearly noting that transition functions of interest are deterministic throughout the paper, (2) Introducing the state-action space as both continuous, along with some discussion about *why* safety is interesting in continuous-but-deterministic environments. For instance, including discussion of consequences to these modelling choices; One interesting consequence is that we are _always_ focused on generalization, as we cannot guarantee the same state will be visited twice. Thus, how does the presence of a hard generalization problem impact safety? To me this question is central.
> >
> > I thank the authors for their careful answers to my other questions and comments. I do believe my concerns have been addressed, and I will increase my score accordingly.

---

> > > ### Author Response · Authors · 2021-08-23
> > > **Response to reviewer hu21**
> > >
> > > We thank the reviewer for taking the time to understand our position and adjust their review accordingly, as well as for offering useful insights regarding the framing of the paper! We agree that assumptions regarding determinism and continuity of the state-action space should be highlighted to a greater degree, and that we should include more discussion of why this is an interesting setting and what challenges remain. We will make these adjustments in the next revision of the paper.

---

### Official Review · Reviewer_Fek3 · 2021-07-21

**Rating:** 8
**Confidence:** 4

**Summary:**

This paper improves sample collection in a model-based exploration scenario, by incorporating a learned reward penalty that prevents unsafe behaviors. The experiments demonstrate improvements in safety during sampling, and sample-efficiency, and policy performance on par with state-of-the-art methods.

**Limitations And Societal Impact:**

The authors have adequately addressed the limitations and potential negative societal impact of their work

**Main Review:**

Model-based policy optimization uses short rollouts from states in the replay buffer to improve the sample efficiency during learning without harming the final policy performance. This paper expands such benefits, to improve the safety of this model-based learning approach, by incorporating a reward penalty over states that are unsafe.

This paper is well-written. The domains are well motivated, and the experimental results demonstrate both improved sample complexity and good final performance. I would like to recommend for acceptance.

That said, it would be good to expand on the number of domains and experiments in the camera-ready of this paper. For example, it would be interesting to see variants of some of the deep mind control suite domains.

**Time Spent Reviewing:**

5 hours split over two days

---

> ### Author Response · Authors · 2021-08-11
> **Response to reviewer Fek3**
>
> We thank the reviewer for the positive review! We are working on adding additional experimental domains based on the ant and humanoid MuJoCo models (which are more challenging than the current tasks), and will have more results to include for the camera-ready.

---

> > ### Comment · Reviewer_Fek3 · 2021-08-18
> > **Response to the Authors**
> >
> > Thanks for mentioning those experiments in progress. Humanoid is indeed more difficult. The reason why I am more interested in results from the DeemMind Control suite is that I found their reward to be better calibrated, so it is easier to tell the performance. I am also speaking under the assumption that switching to DMC could be a trivial exercise at this point given the availability of wrappers such as gym-dmc.

---

> > > ### Author Response · Authors · 2021-08-23
> > > **Response to reviewer Fek3**
> > >
> > > Thank you for the pointer to gym-dmc! We will look into the feasibility of adding additional experiments with the DeepMind Control suite for the camera-ready. (It is not entirely trivial, as a reasonable safety constraint must still be defined, but I think it is reasonably likely we can get it to work. The bigger difficulty is the computational requirements to run hyperparameter searches for both our algorithm and the baseline algorithms.)

---

### Decision · Program_Chairs · 2021-09-27

**Decision:**

Accept (Poster)

**Comment:**

This paper presents a model-based policy optimization method that reduces the frequency that unsafe states are visited. All three reviewers recommend acceptance, one strongly. The primary concerns of the reviewers centered around whether the proposed method would extend to interesting problem settings. After the discussion, the reviewers were all convinced that the covered settings are of interest, and the extension to even more settings (stochastic transitions) may be feasible as an avenue for future work. The AC recommends that the authors work to improve the clarity of these points in the paper to ensure that future readers do not run into the same points of confusion that tripped up the reviewers.